# Predictive symptoms for COVID-19 in the community: REACT-1 study of over 1 million people

Joshua Elliott[1,2,3☉], Matthew Whitaker[1,2☉], Barbara Bodinier[1,2☉], Oliver Eales[4,5], Steven Riley[4,5], Helen Ward[1,6,7], Graham Cooke[6,7,8], Ara Darzi[6,7,9], Marc Chadeau-Hyam[1,2,10‡*], Paul Elliott[1,2,6,7,10,11‡*]

**1** Department of Epidemiology and Biostatistics, School of Public Health, Imperial College London, London, United Kingdom, **2** MRC Centre for Environment and Health, Imperial College London, London, United Kingdom, **3** Royal Surrey NHS Foundation Trust, Guildford, United Kingdom, **4** MRC Centre for Global infectious Disease Analysis, Imperial College London, London, United Kingdom, **5** Abdul Latif Jameel Institute for Disease and Emergency Analytics, Imperial College London, London, United Kingdom, **6** Imperial College Healthcare NHS Trust, London, United Kingdom, **7** National Institute for Health Research Imperial Biomedical Research Centre, London, United Kingdom, **8** Department of Infectious Disease, Imperial College London, London, United Kingdom, **9** Institute of Global Health Innovation, Imperial College London, London, United Kingdom, **10** Health Data Research UK London, Imperial College London, London, United Kingdom, **11** UK Dementia Research Institute, Imperial College London, London, United Kingdom

☉ These authors contributed equally to this work.
‡ These authors are joint senior authors on this work.
* m.chadeau@imperial.ac.uk (MCH); p.elliott@imperial.ac.uk (PE)

**Data Availability Statement:** Access to REACT-1 data is restricted to protect participants' anonymity. Researchers wishing to inquire about access to data should email react.access@imperial.ac.uk.

## Abstract

### Background

Rapid detection, isolation, and contact tracing of community COVID-19 cases are essential measures to limit the community spread of severe acute respiratory syndrome coronavirus 2 (SARS-CoV-2). We aimed to identify a parsimonious set of symptoms that jointly predict COVID-19 and investigated whether predictive symptoms differ between the B.1.1.7 (Alpha) lineage (predominating as of April 2021 in the US, UK, and elsewhere) and wild type.

### Methods and findings

We obtained throat and nose swabs with valid SARS-CoV-2 PCR test results from 1,147,370 volunteers aged 5 years and above (6,450 positive cases) in the REal-time Assessment of Community Transmission-1 (REACT-1) study. This study involved repeated community-based random surveys of prevalence in England (study rounds 2 to 8, June 2020 to January 2021, response rates 22%–27%). Participants were asked about symptoms occurring in the week prior to testing. Viral genome sequencing was carried out for PCR-positive samples with N-gene cycle threshold value < 34 ($N$ = 1,079) in round 8 (January 2021). In univariate analysis, all 26 surveyed symptoms were associated with PCR positivity compared with non-symptomatic people. Stability selection (1,000 penalized logistic regression models with 50% subsampling) among people reporting at least 1 symptom identified 7 symptoms as jointly and positively predictive of PCR positivity in rounds 2–7 (June to December 2020): loss or change of sense of smell, loss or change of sense of taste, fever,

Summary statistics and descriptive tables from the current REACT-1 study are available in the Supplementary Information and at the following link https://github.com/mrc-ide/reactidd/tree/master/inst/extdata/symptoms_prediction_paper Source codes used for the present analyses are available at https://github.com/mrc-ide/reactidd/tree/master/R/symptom_prediction_paper_scripts Additional summary statistics and results from the REACT-1 programme are also available at: https://www.imperial.ac.uk/medicine/research-and-impact/groups/react-study/real-time-assessment-of-community-transmission-findings/ and https://github.com/mrc-ide/reactidd/tree/master/inst/extdata REACT-1 Study Materials are available for each round at https://www.imperial.ac.uk/medicine/research-and-impact/groups/react-study/react-1-study-materials/.

**Funding:** This work was funded by the Department of Health and Social Care in England. MC-H and MW acknowledge support from the H2020-EXPANSE project (Horizon 2020 grant No 874627). MC-H, JE, and BB acknowledge support from Cancer Research UK, Population Research Committee Project grant 'Mechanomics' (grant No 22184 to MC-H). HW is a NIHR Senior Investigator and acknowledges support from NIHR Biomedical Research Centre of Imperial College NHS Trust, NIHR School of Public Health Research, NIHR Applied Research Collaborative North West London, Wellcome Trust (UNS32973). SR acknowledges support from MRC Centre for Global Infectious Disease Analysis, National Institute for Health Research (NIHR) Health Protection Research Unit (HPRU), Wellcome Trust (200861/Z/16/Z, 200187/Z/15/Z), and Centres for Disease Control and Prevention (US, 442 U01CK0005-01-02). GC is supported by an NIHR Professorship. PE is Director of the MRC Centre for Environment and Health (MR/L01341X/1, MR/S019669/1), and BB received a studentship from this Centre. PE acknowledges support from the National Institute for Health Research Imperial Biomedical Research Centre and the NIHR Health Protection Research Units in Chemical and Radiation Threats and Hazards, and in Environmental Exposures and Health, the British Heart Foundation (BHF) Centre for Research Excellence at Imperial College London (RE/18/4/34215) and the UK Dementia Research Institute at Imperial (MC_PC_17114). We thank The Huo Family Foundation for their support of our work on COVID-19. The funders had no role in study design, data collection and analysis, decision to publish, or preparation of the manuscript.

**Competing interests:** I have read the journal's policy and the authors of this manuscript have the

new persistent cough, chills, appetite loss, and muscle aches. The resulting model (rounds 2–7) predicted PCR positivity in round 8 with area under the curve (AUC) of 0.77. The same 7 symptoms were selected as jointly predictive of B.1.1.7 infection in round 8, although when comparing B.1.1.7 with wild type, new persistent cough and sore throat were more predictive of B.1.1.7 infection while loss or change of sense of smell was more predictive of the wild type. The main limitations of our study are (i) potential participation bias despite random sampling of named individuals from the National Health Service register and weighting designed to achieve a representative sample of the population of England and (ii) the necessary reliance on self-reported symptoms, which may be prone to recall bias and may therefore lead to biased estimates of symptom prevalence in England.

## Conclusions

Where testing capacity is limited, it is important to use tests in the most efficient way possible. We identified a set of 7 symptoms that, when considered together, maximize detection of COVID-19 in the community, including infection with the B.1.1.7 lineage.

## Author summary

### Why was this study done?

- The rapid detection of SARS-CoV-2 infection in the community is key to ensuring efficient control of transmission via isolation.

- Eligibility for community PCR testing is determined based on the reported presence of several (predetermined) symptoms, which may vary from one country to another.

- Quantitative evidence measuring which symptoms are the most informative of a COVID-19 infection remains scarce.

### What did the researchers do and find?

- Data were collected from over 1 million participants in the REACT-1 study (June 2020 to January 2021), for whom 26 symptoms were assayed and the results of a PCR test were available.

- Adopting a variable selection approach, we sought to determine the best combination of symptoms jointly and complementarily predictive of PCR positivity and investigated whether these symptoms were the same between individuals infected by the wild-type virus and those infected by the B.1.1.7 variant.

- We identified 7 symptoms that were jointly predictive of PCR positivity and appeared to vary only marginally across age groups: loss or change of sense of smell, loss or change of sense of taste, fever, new persistent cough, chills, appetite loss, and muscle aches.

- These symptoms were also predictive of the B.1.1.7 infection, together with sore throat (to a lesser extent).

following competing interests: PE is the director of
the MRC Centre of Environment and Health (MR/
L01341X/1 and MC/S019669/1) and has no
conflict of interest to disclose. M C-H holds shares
in the O-SMOSE company and has no conflict of
interest to disclose. Consulting activities conducted
by the company are independent of the present
work. All other authors have no conflict of interest
to disclose.

**Abbreviations:** AUC, area under the curve; LASSO,
least absolute shrinkage and selection operator;
LTLA, lower-tier local authority; REACT-1, REal-
time Assessment of Community Transmission-1;
SARS-CoV-2, severe acute respiratory syndrome
coronavirus 2.

## What do these findings mean?

- Taken together, these 7 symptoms can improve the detection of COVID-19 infection in the community.

- Using this sparse set of symptoms for test allocation would increase the number of tests performed (up to 30%–40% of symptomatic individuals being tested) but would enable up to 75% of symptomatic cases to be detected.

- This set of 7 symptoms is also predictive of B.1.1.7 infection and performs similarly across age groups. Its use would maximize the case detection rate in the community and would be particularly relevant in situations where test capacity is limited.

## Introduction

To control the severe acute respiratory syndrome coronavirus 2 (SARS-CoV-2) pandemic, rapid identification and isolation of infected individuals is essential [1,2], together with testing and isolation of their contacts [3,4]. A range of symptoms have been identified as associated with COVID-19. According to the US Centers for Disease Control and Prevention, these include fever or chills, cough, shortness of breath or difficulty breathing, fatigue, muscle or body aches, headache, new loss of taste or smell, sore throat, congestion or runny nose, nausea or vomiting, and diarrhea [5]. However, it is unclear which symptoms are the most informative of a COVID-19 diagnosis. This is important in settings where test supply is limited.

A novel SARS-CoV-2 variant of concern, VOC 20DEC-01 (lineage B.1.1.7), was first identified in England in September 2020 and became the dominant lineage in the UK within 4 months [6]. As of April 2021, it had been detected in 114 countries [7] and had become the dominant lineage in the US, Europe, and elsewhere [7,8].

Here, we used data from the REal-time Assessment of Community Transmission-1 (REACT-1) study to identify a parsimonious set of symptoms that, taken together, are the most predictive of SARS-CoV-2 PCR positivity. For data collected during January 2021 we also compare predictive symptoms for B.1.1.7 infection versus wild type, identified via viral genome sequencing.

## Methods

### Study population

REACT-1 is a series of community prevalence surveys of SARS-CoV-2 virus swab positivity in England, conducted at approximately monthly intervals since May 2020. Using the National Health Service patient register across the 315 lower-tier local authorities (LTLAs) in England, recruitment letters were sent to a random nationally representative sample of individuals aged 5 years and over, with a separate (non-overlapping) sample selected at each round. We therefore approached a different base sample at each round and did not have repeat samples to account for. Participant sampling aimed to achieve approximately equal numbers of participants in each LTLA. Random samples were drawn stratified by LTLA, with larger numbers selected in some LTLAs to address variable response rates at the LTLA level. Up to 160,000 valid responses and viable swabs were obtained at each round [9]. Participation involved a self-administered throat and nasal swab, and the completion of a short online or telephone

questionnaire including information on demographic variables, household composition, behaviors, and recent symptoms. The questionnaires used are available on the study website (https://www.imperial.ac.uk/medicine/research-and-impact/groups/react-study/react-1-study-materials/). At the time of survey completion, participants were unaware of the result of their swab test. In the survey, participants were asked about new symptoms occurring in the week preceding the swab. These included a set of 26 clinically relevant symptoms potentially related to COVID-19: (i) loss or change of sense of smell and loss or change of sense of taste, (ii) coryzal symptoms (runny nose, sneezing, blocked nose, sore eyes, sore throat, and hoarse voice), (iii) gastrointestinal symptoms (appetite loss, nausea/vomiting, diarrhea, and abdominal pain/belly ache), (iv) fatigue-related symptoms (tiredness, severe fatigue, heavy arms/legs, and difficulty sleeping), (v) respiratory and cardiac symptoms (new persistent cough, shortness of breath, chest pain, and tight chest), and (vi) other flu-like and miscellaneous symptoms (fever, muscle aches, chills, headache, dizziness, and numbness/tingling). Data from round 1 were excluded as the symptom questions asked in that round were not consistent with those in the subsequent rounds [9].

Here, we use REACT-1 data from rounds 2 to 7 (June to December 2020) to identify a parsimonious set of symptoms (from 1 week prior to testing) that are jointly predictive of SARS-CoV-2 PCR positivity and assess their performance among holdout community population-based data from rounds 2 to 7 and, separately, round 8 (January 2021).

PCR-positive swab samples from round 8 with N-gene cycle threshold value < 34 and sufficient sample volume underwent genome sequencing. Viral RNA was amplified using the ARTIC protocol [10], and sequencing libraries were prepared with CoronaHiT [11]. Samples were sequenced using the Illumina NextSeq 500 platform. Each run included 1 positive and 1 negative control per 96 samples. Sequencing data were analyzed using the ARTIC bioinformatic pipeline [12]. Lineages were assigned using PangoLEARN [13].

## Statistical analyses

Twenty-five participants (19 in rounds 2–7 and 9 in round 8) were excluded due to missing information on sex. We used univariate logistic regression to model the risk of testing positive for SARS-CoV-2 as a function of symptoms reported in the week prior to testing. For multivariable models, round 2–7 data (restricted to people reporting any of the 26 surveyed symptoms) were split into a 70% training set (76,187 observations, of which 1,078 were positive) and a 30% test set (32,651 observations, of which 461 were positive). We adopted a variable selection approach with stability selection using least absolute shrinkage and selection operator (LASSO) penalized logistic regression with all 26 surveyed symptoms as predictors and PCR positivity as the outcome [14]. LASSO models were fit on 1,000 independent random 50% subsamples of the 70% training set. The stability selection penalty parameter was calibrated to give a per family error rate of fewer than 5 falsely selected symptoms [14,15]. The proportion of penalized models where each symptom was included (its selection proportion) was used as a measure of its importance; those with selection proportion above 50% were considered stably selected and were included in a predictive model of PCR positivity, where mean penalized odds ratios across models (where selected) were used as weightings. The resultant model was then applied to the 30% holdout test set (rounds 2–7) and, separately, to round 8 data. We then estimated the proportion of symptomatic COVID-19 cases that would be detected using this predictive model across all levels of symptomatic community testing.

As a series of sensitivity analyses, we used the same univariate and multivariate approaches to (i) investigate age-specific symptoms by stratifying our study population into 3 age groups (5 to 17 years, 18 to 54 years, and 55 years old or older); (ii) account for possibly differential

predictive abilities of symptoms in relation to their sequence/timing of onset, by restricting the list of symptoms to those reported first (rather than any symptom reported in the week prior to testing); and (iii) search for (sets of) symptoms potentially discriminating B.1.1.7 versus wild-type infections among test-positive cases.

All calculations were done with the R computational environment, version 4.0.2, using the logistic LASSO algorithm as implemented in the glmnet package and in-house scripts for the stability selection (available at https://github.com/mrc-ide/reactidd/tree/master/R/symptom_prediction_paper_scripts).

## Ethical approval

We obtained research ethics approval from the South Central–Berkshire B Research Ethics Committee (IRAS ID: 283787).

## Patient and public involvement

Participants in the REACT-1 study were not involved in the definition of our research question or in the outcome measurements. They were not involved in developing the analytical plan or the implementation of the study. There are no plans to disseminate the results of the research directly to the study participants or the relevant patient community. However, a public advisory panel provides regular review of the study processes and results, and links to published reports from the study are available on the study website (https://www.imperial.ac.uk/medicine/research-and-impact/groups/react-study/real-time-assessment-of-community-transmission-findings/).

## Results

### Descriptive and univariate analyses

The key characteristics of the study population in rounds 2–7 and 8 for the full sample and for those reporting symptoms are summarized in S1 Table. Of the 979,709 respondents (after 19 exclusions) with a valid swab test in REACT-1 rounds 2–7, 870,872 (88.9%) reported no symptoms in the week prior to testing while 108,837 (11.1%) reported at least 1 of the 26 surveyed symptoms. We detected 4,168 PCR-positive cases, of whom 1,538 (36.9%) reported 1 or more symptoms in the past week. In round 8, there were 167,636 participants (after 6 exclusions), of whom 146,701 (87.5%) reported no symptoms in the past week and 20,935 (12.5%) reported at least 1 of the 26 surveyed symptoms. We detected 2,282 PCR-positive cases, of whom 1,031 (45.1%) reported 1 or more symptoms in the past week (Fig 1; S2 Table). In univariate analyses, each of the 26 surveyed symptoms was associated with PCR positivity in rounds 2–7 and round 8 (Fig 2; S2 Table). In rounds 2–7, with a mean prevalence of PCR positivity of 0.46%, the positive predictive value for any symptom was 1.4%, with an odds ratio for PCR positivity of 4.7 compared with non-symptomatic individuals. In round 8, with a PCR positivity prevalence of 1.36%, the positive predictive value for any symptom was 4.9%, with an odds ratio for PCR positivity of 6.0 compared with non-symptomatic individuals (Fig 1).

### Multivariable analyses and community case detection

Seven symptoms were selected as jointly positively predictive of PCR positivity in a LASSO stability selection model trained on round 2–7 data: loss or change of sense of smell, loss or change of sense of taste, fever, new persistent cough, chills, appetite loss, and muscle aches. In addition, numbness/tingling was selected as jointly but negatively predictive of PCR positivity. The resultant model gave an area under the curve (AUC) of 0.75 for holdout round 2–7 test

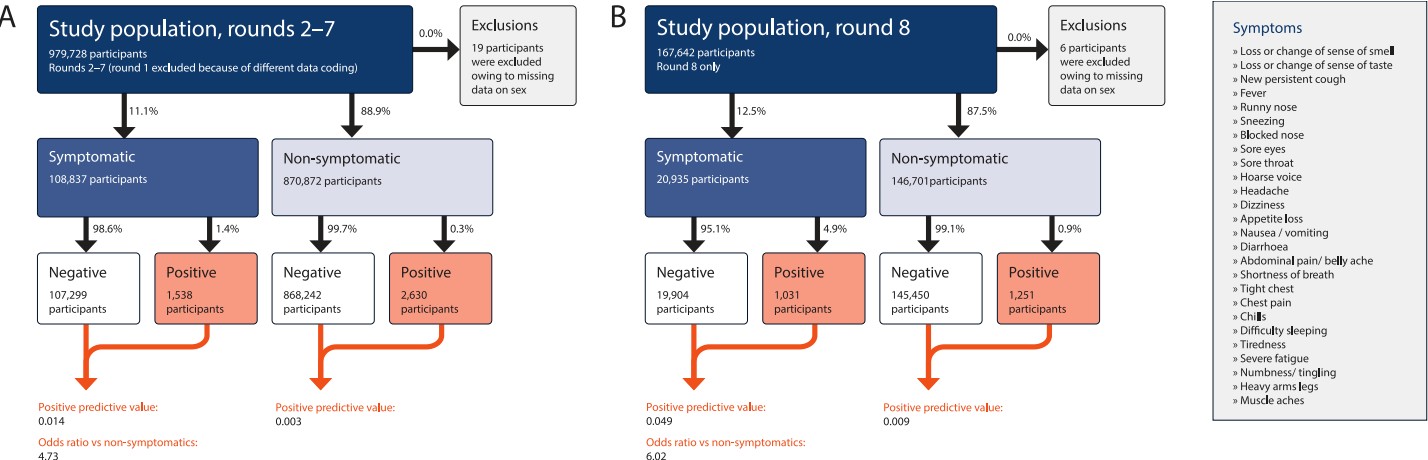

**Fig 1. Flow chart showing numbers of participants by symptom status and PCR result.** (A) Rounds 2–7 and (B) round 8 of the REACT-1 study.

data and 0.77 for round 8 data, with no improvement in AUC upon inclusion of other symptoms (Fig 3). Testing people in the community with at least 1 of the 7 selected positively predictive symptoms gave sensitivity, specificity, and positive predictive value of 71%, 64%, and 1.9% in holdout round 2–7 test data and 74%, 64%, and 9.7% in round 8 data, respectively. Performance of the stability selection model at varying levels of community testing is shown in S1 Fig; for example, testing 10% of symptomatic individuals would detect around half of all symptomatic cases.

## B.1.1.7 (Alpha) lineage versus wild-type symptoms

Of the 2,282 positive cases in round 8, 1,088 had cycle threshold value < 34 and underwent viral genome sequencing; 898 (82.5%) were B.1.1.7, 181 (16.6%) were wild type, and 8 were other lineages. Characteristics of the positive cases with B.1.1.7 and wild type in round 8 were similar except for the region of the case, with a marked excess of B.1.1.7 cases in East of England, South East, and London (S3 Table). LASSO stability selection models fit on round 8 data with an outcome of B.1.1.7 versus PCR negative selected the same 7 positively predictive symptoms as in rounds 2–7 (Fig 4). In univariate analysis within round 8, there was slightly higher reporting of any symptom for B.1.1.7 infection versus wild type. Specifically, we found a higher prevalence of sore throat, new persistent cough, fever, difficulty sleeping, dizziness, and nausea/vomiting at a nominal 0.05 significance level (Fig 5A). Stability selection models identified sore throat and new persistent cough as jointly and positively associated with B.1.1.7 infection compared with wild-type infection, while loss or change of sense of smell was selected as negatively associated, i.e., more predictive of wild type infection (Fig 5B).

## Sensitivity analyses

In age-stratified analyses of round 2–7 data, 6 (5–17 years) or 7 (18–54 and 55+ years) symptoms were jointly selected. New persistent cough was not selected for those aged 5–17 years but was for adults, while chills, fever, loss or change of sense of smell, and loss or change of sense of taste were selected in all age groups. Additional age-specific selected symptoms (positive coefficients) were headache (5–17 years), appetite loss (18–54 and 55+ years), and muscle aches (18–54 years), and age-specific negatively associated symptoms (negative coefficients) were runny nose (5–17 years) and numbness/tingling (55+ years) (S2 Fig).

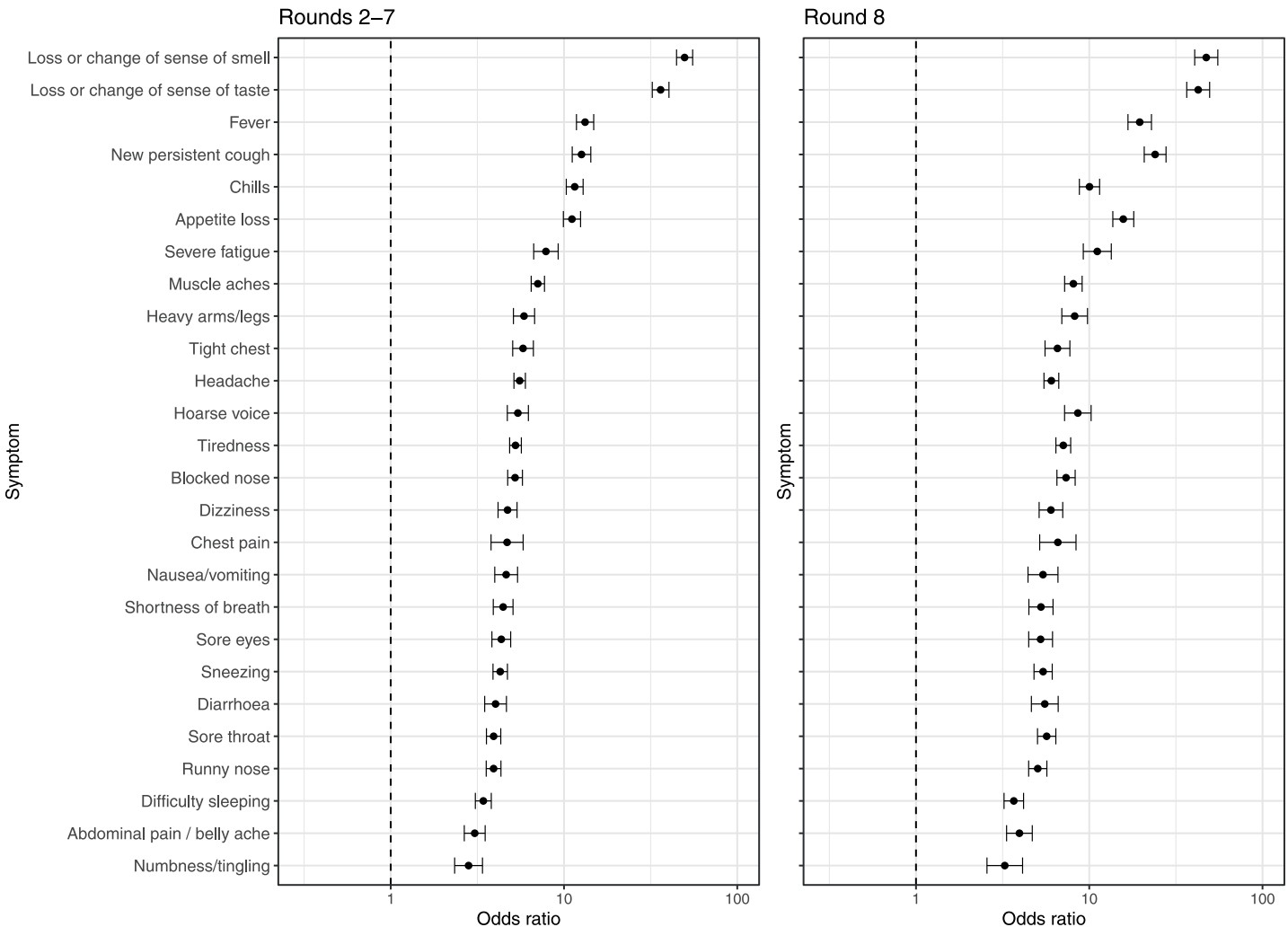

Rounds 2–7 · Round 8

**Fig 2. Results from univariate logistic regression models of PCR positivity for 26 surveyed symptoms.** Effect size estimates are expressed as odds ratios (95% confidence intervals) in rounds 2–7 (left) and round 8 (right).

Considering first reported symptom, instead of any symptom, as the predictor, in univariate analysis, all first reported symptoms among the 26 surveyed were associated with PCR positivity in both round 2–7 and round 8 data (S3 Fig). In multivariable analysis, LASSO stability selection on round 2–7 data included headache as positively predictive of PCR positivity instead of appetite loss (S4 Fig); otherwise, the selected (positively associated) symptom set was the same as in Fig 3, although with lower predictive performance (AUC of 0.68 and 0.66 for holdout round 2–7 test data and round 8 data, respectively).

## Discussion

In this study of over 1 million people in England, we found that 7 symptoms stably and jointly predicted SARS-CoV-2 PCR positivity; these were loss or change of sense of smell, loss or change of sense of taste, fever, new persistent cough, chills, appetite loss, and muscle aches. The first 4 of these symptoms are currently used in the UK to determine eligibility for community PCR testing, selected at a time (May 2020) when testing capacity was limited. Based on our findings, this symptom set is too restrictive, and in order to improve PCR positivity

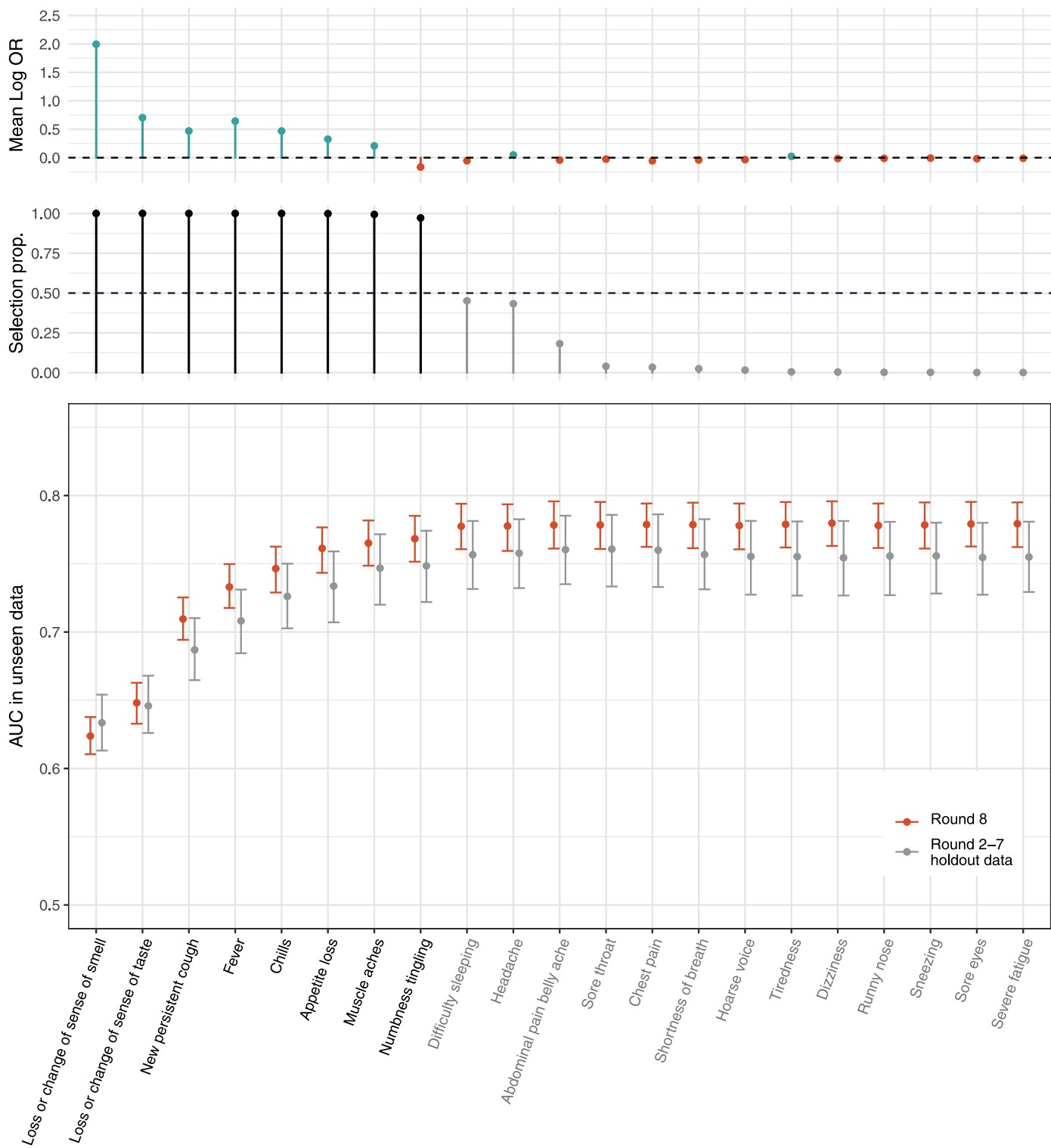

**Fig 3. Selected symptoms predictive of COVID-19.** Results of LASSO stability selection using 1,000 models (with 50% subsamples of training data from rounds 2–7). Mean (penalized) log odds ratios (log ORs) across all models are shown in the top panel. Positive regression coefficients are presented in teal, and negative in red. Only symptoms selected at least once are displayed. The selection proportions (selection prop.; proportion of 1,000 models that included each symptom) are shown in the middle panel; the horizontal dashed line shows the selection threshold of 50%. Symptoms are ordered according to their selection proportions, and selected symptoms are in black. The bottom panel shows the area under the curve (AUC) of models adding each variable in order of selection proportion (from left to right) in both holdout data from rounds 2–7 (grey) and data from round 8 (red).

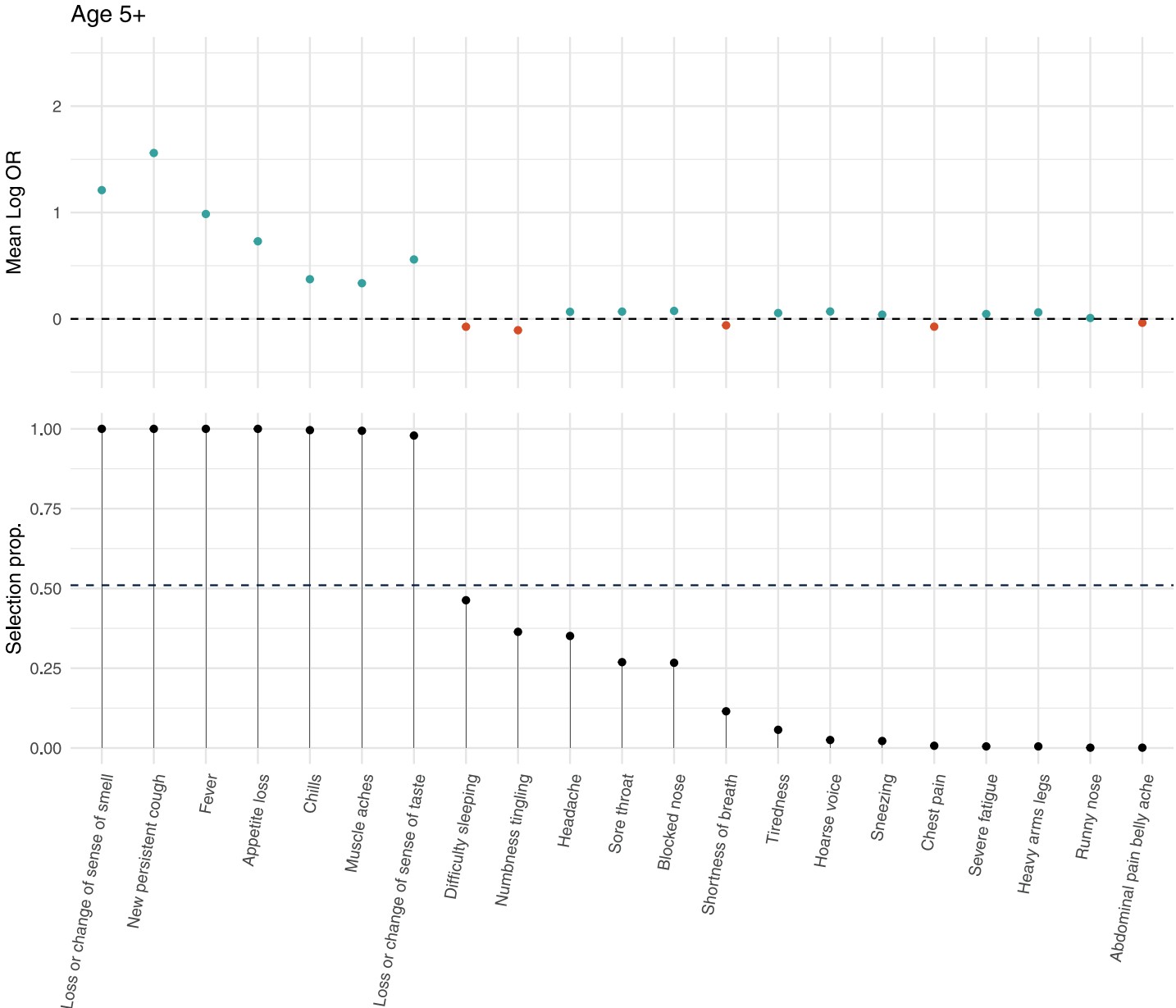

**Fig 4. Symptoms predictive of B.1.1.7 infection.** LASSO stability selection for symptoms predictive of B.1.1.7 (Alpha) lineage infection versus symptomatic people (aged 5+ years) testing PCR negative in round 8. Mean log odds ratio (Log OR) and selection proportion (selection prop.) are represented for each symptom in the top and bottom panels, respectively. Positive regression coefficients are presented in teal, and negative in red.

detection rates and consequently improve control of viral transmission via isolation measures, we would propose to extend the list of symptoms used for triage to all 7 symptoms we identified. This approach would have the advantage of increasing the yield of detected cases, leading to greater numbers of infected people being required to self-isolate, thereby reducing the pool of infection in the community.

The use of the 7 symptoms we identified for PCR test allocation would result in 30% to 40% of symptomatic individuals in England being eligible for a test (versus 10% currently) and, if all those eligible were tested, would result in the detection of 70% to 75% of the positive cases.

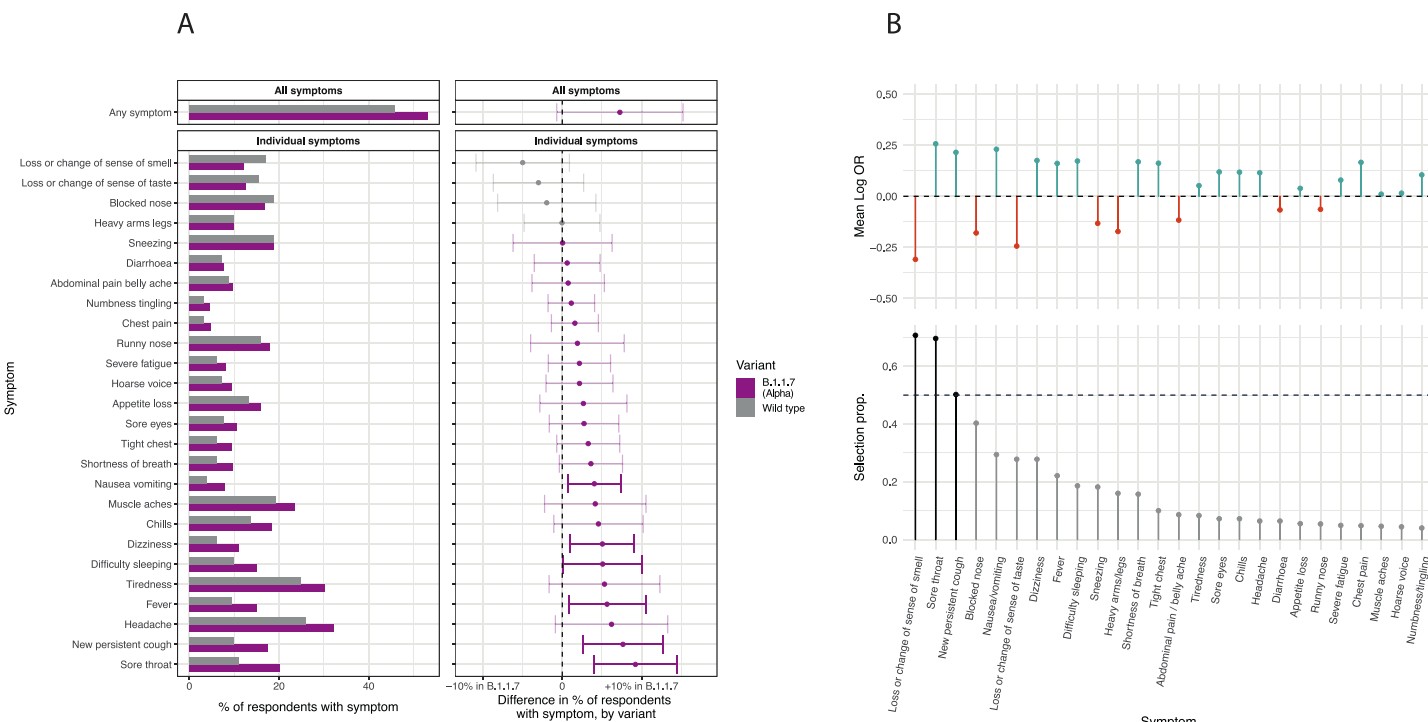

**Fig 5. B.1.1.7 versus wild-type symptoms.** Comparison of symptom profile in B.1.1.7 (Alpha) lineage versus wild-type infection among 1,124 people testing positive in round 8 (other lineages excluded, *N* = 8). (A) Proportion of people reporting each symptom by lineage (left panel), and the differences in proportions with 95% confidence intervals (right panel). (B) Results of LASSO stability selection (using 1,000 models with 50% subsampling) with B.1.1.7 infection, versus wild-type infection as the outcome, summarized by the mean log odds ratio (Log OR) and selection proportion (selection prop.) for each of the symptoms selected at least once. Positive regression coefficients are presented in teal, and negative in red. The horizontal dashed line represents the selection threshold of 50%.

Further extending the list of symptoms for test allocation would increase the number of tests performed (up to 3-fold) and potentially result in a 100% positive detection rate if all symptomatic individuals were tested, but with increasing prevalence rates would exceed testing capacity. We believe that our approach, relying on these 7 symptoms, provides a reasonable balance between number of tests performed and detection rates in England, but alternative approaches could be envisaged depending on national population size, composition, and resources. In particular, testing individuals reporting any COVID-19-related symptoms has been implemented in some countries that have adopted a zero COVID-19 policy (e.g., Australia and New Zealand). Such investments have been justified by arguing that, if successful, this approach leads to an earlier and more efficient control of transmission of SARS-CoV-2, in turn reducing the prevalence of the infection and hence of symptoms, thus reducing the testing capacity needs.

Irrespective of the test allocation strategy, prevention efficacy strongly relies on the accurate reporting of symptoms by individuals in the general population. To ensure adherence to self-isolation measures and lessen the burden of such measures in socially disadvantaged individuals, efficient financial and other support policies should be encouraged. Failure to implement such targeted policies could introduce a social gradient in the willingness to report symptoms [16] and attend for testing, which in turn could lead to increased transmission of SARS-CoV-2 in more deprived populations, as has been observed in England [17].

Other studies investigating sets of symptoms predictive of PCR positivity have been published [18]. As in our study, these identified loss or change of sense of smell, loss or change of sense of taste, fever, and new persistent cough as consistent predictors of infection. Of the 3

additional symptoms we identify (chills, appetite loss, and muscle aches), appetite loss was found previously to be associated with PCR positivity in the UK [18]. Unlike other studies, REACT-1 involves community-based random samples of individuals in the population, and with over 1 million participants, it provides reliable, reproducible, and representative estimates of prevalence and prediction of PCR positivity in England from a combination of informative symptoms.

The second wave of COVID-19 in England coincided with the emergence of VOC 20DEC-01 (lineage B.1.1.7), which became dominant in the UK by January 2021 [6] and subsequently in many countries around the world as of April 2021 [7]. B.1.1.7 is defined by 17 mutations; 8 of these affect the viral spike protein, the means by which SARS-CoV-2 binds to angiotensin-converting enzyme 2 (ACE2) and enters host cells. These spike mutations might confer an evolutionary benefit. For example, the spike deletion ΔH69/ΔV70 enhances viral infectivity in vitro [19], and N501Y may enhance spike binding affinity to ACE2 [20]. From modeling, it has been estimated that B.1.1.7 is 40% to 90% more transmissible than earlier lineages [21] and might be associated with an increased risk of hospitalization and death [22,23].

Here we show that the same symptoms were selected as jointly predictive of B.1.1.7 infection as for earlier lineages. However, when we compared B.1.1.7 with wild-type infections, new persistent cough and sore throat were jointly selected as predictive of B.1.1.7 infections, while loss or change of sense of smell was predictive of wild-type infections. This is consistent with findings from the UK Office for National Statistics Coronavirus (COVID-19) Infection Survey, where people testing PCR positive for lineages compatible with B.1.1.7 were less likely to report loss or change of sense of taste or smell (both symptoms combined), and more likely to report cough, compared with non-B.1.1.7-compatible lineages [24].

Our study has limitations. First, it is uncertain to what extent our findings are generalizable to other settings. However, our sampling procedure was designed to ensure good representation across the whole population in England, including capturing sociodemographic and ethnic diversity. Second, our data relied on a time-resolved series of reported symptoms from self-administered questionnaires. These may be subject to recall bias and may not precisely represent the individual dynamics of symptom onset. However, our self-reported data are based on representative community-based samples and may not share the same limitations as the routine reporting of symptoms on which test allocation and isolation measures are based [25]. Furthermore, participants were unaware of their test results at the time of symptom report, which would limit reporting and information bias. Third, as sampling in each round was cross-sectional, some individuals may have been infected (and had symptoms) more than 1 week before the swab was obtained but were no longer symptomatic at the time of the study or might have gone on to develop symptoms after testing. Fourth, despite our careful sample weighting to account for possible variations in response rates by sex, age, region, deprivation, and ethnicity, we cannot rule out residual selection bias. Finally, a number of B.1.1.7 infections may have been included in PCR positive samples from rounds 5 to 7 (estimated proportions of B.1.1.7 infection among all SARS-CoV-2 infections in England are 0.7% [95% CI 0.6%, 0.9%] and 10.6% [95% CI 10.1%, 11.2%] for November [round 6] and December [round 7], respectively; S5 Fig) [26]. Overall, B.1.1.7 would represent less than 4% of the PCR-positive cases in rounds 2–7, and because our results were suggestive of a consistent symptomatology for both B.1.1.7 and wild-type infections, possible confounding of our results by the inclusion of early B.1.1.7 incident cases is limited.

Our main analyses fit models to the population at all ages. In age-stratified analyses, however, we found a different symptom profile among children and adolescents (ages 5–17 years), where headache (positive association) and runny nose (negative association) were selected, but new persistent cough was not. This may have implications for symptomatic testing in

school-aged children. Headache was also selected (instead of appetite loss) for first reported symptom at all ages (5+ years), but a model derived from first reported symptom was less predictive of PCR positivity than the model based on all symptoms within the week prior to testing.

In summary, we show that using a combination of 7 symptoms to determine test eligibility would maximize the case detection rate in the community under testing capacity constraints such as those faced in England between June 2020 and January 2021. This has policy relevance for countries where there is limited testing capacity. We identified the same symptom set for predicting B.1.1.7, which by April 2021 had become the predominant lineage in the UK, US, and many other countries worldwide.

## Supporting information

**S1 Fig. Univariate effect of each symptom reported in the week prior to testing on risk of PCR positivity in rounds 2–7 and round 8.**
(DOCX)

**S2 Fig. Age-stratified LASSO stability selection on testing PCR positive.**
(DOCX)

**S3 Fig. Univariate effect of first reported symptom on risk of PCR positivity in rounds 2–7 and round 8.**
(DOCX)

**S4 Fig. LASSO stability selection for first reported symptom.**
(DOCX)

**S5 Fig. Proportion of B.1.1.7 variant cases among all cases in England, October 2020–January 2021.**
(DOCX)

**S1 Supplementary Methods. Supplementary methods.**
(DOCX)

**S1 Table. Key characteristics of the REACT-1 study population for rounds 2–7 and round 8.** Results are presented for the full study population.
(DOCX)

**S2 Table. Descriptive statistics for symptoms and SARS-CoV-2 PCR test results among REACT-1 participants, rounds 2–7 and round 8.**
(DOCX)

**S3 Table. Characteristics of the round 8 PCR-positive participants in whom lineage data were available.** Numbers are presented for cases with wild-type ($N$ = 181) and B.1.1.7 ($N$ = 898) infections. The statistical significance of differences between these 2 groups is evaluated using a chi-squared test for categorical variables. We report the *p*-value for the null hypothesis of no difference.
(DOCX)

## Acknowledgments

We thank key collaborators on this work—Ipsos MORI: Kelly Beaver, Sam Clemens, Gary Welch, Nicholas Gilby, Kelly Ward, and Kevin Pickering; Institute of Global Health Innovation at Imperial College London: Gianluca Fontana, Sutha Satkunarajah, Didi Thompson,

Justine Alford, and Lenny Naar; Molecular Diagnostic Unit, Imperial College London: Prof. Graham Taylor; North West London Pathology and Public Health England for help in calibration of the laboratory analyses; Quadram Institute, Norwich, UK: Andrew Page and Justin O'Grady, who carried out the sequencing and annotation; NHS Digital for access to the National Health Service register; and the Department of Health and Social Care for logistic support.

## Author Contributions

**Conceptualization:** Joshua Elliott, Steven Riley, Graham Cooke, Ara Darzi, Marc Chadeau-Hyam, Paul Elliott.

**Data curation:** Matthew Whitaker, Barbara Bodinier, Oliver Eales.

**Formal analysis:** Joshua Elliott, Matthew Whitaker, Barbara Bodinier, Oliver Eales, Marc Chadeau-Hyam.

**Funding acquisition:** Paul Elliott.

**Investigation:** Joshua Elliott, Matthew Whitaker, Barbara Bodinier, Graham Cooke, Ara Darzi, Marc Chadeau-Hyam, Paul Elliott.

**Methodology:** Joshua Elliott, Matthew Whitaker, Barbara Bodinier, Steven Riley, Helen Ward, Graham Cooke, Ara Darzi, Marc Chadeau-Hyam, Paul Elliott.

**Software:** Barbara Bodinier.

**Supervision:** Steven Riley, Helen Ward, Marc Chadeau-Hyam, Paul Elliott.

**Validation:** Oliver Eales.

**Visualization:** Joshua Elliott, Matthew Whitaker, Barbara Bodinier.

**Writing – original draft:** Joshua Elliott, Matthew Whitaker, Barbara Bodinier, Oliver Eales, Steven Riley, Helen Ward, Graham Cooke, Marc Chadeau-Hyam, Paul Elliott.

**Writing – review & editing:** Joshua Elliott, Matthew Whitaker, Helen Ward, Graham Cooke, Ara Darzi, Marc Chadeau-Hyam, Paul Elliott.

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
