## [Editor Report · Decision Letter 0]

12 May 2021

Dear Dr Chadeau-Hyam, 

Thank you for submitting your manuscript entitled "Predictive symptoms for COVID-19 in the community: REACT-1 study of over one million people" for consideration by PLOS Medicine.

Your manuscript has now been evaluated by the PLOS Medicine editorial staff and I am writing to let you know that we would like to send your submission out for external peer review.

Please re-submit your manuscript within two working days, i.e. by May 14 2021 11:59PM.

Kind regards,

Callam Davidson

Associate Editor

PLOS Medicine

---

## [Decision Letter · Decision Letter 1]

1 Jul 2021

Dear Dr. Chadeau-Hyam,

Thank you very much for submitting your manuscript "Predictive symptoms for COVID-19 in the community: REACT-1 study of over one million people" (PMEDICINE-D-21-02112R1) for consideration at PLOS Medicine. 

Your paper was discussed among all the editors here. It was also discussed with an academic editor with relevant expertise, and sent to independent reviewers, including a statistical reviewer. The reviews are appended at the bottom of this email and any accompanying reviewer attachments can be seen via the link below:

[LINK]

In light of these reviews, we will not be able to accept the manuscript for publication in the journal in its current form, but we would like to consider a revised version that addresses the reviewers' and editors' comments fully. You will appreciate that we cannot make a decision about publication until we have seen the revised manuscript and your response, and we expect to seek re-review by one or more of the reviewers. 

We hope to receive your revised manuscript by Jul 22 2021 11:59PM. Please email us (plosmedicine@plos.org) if you have any questions or concerns.

Please let me know if you have any questions, and we look forward to receiving your revised manuscript. 

Sincerely,

Callam Davidson, 

Associate Editor

PLOS Medicine

plosmedicine.org

Please update the final sentence of the ‘Methods and Findings’ section of your abstract to list 2-3 of the study’s main limitations (the sentence should begin ‘Limitations of this study include…’ or similar). 

Please ensure you have provided all the necessary information that would allow a reader to access the data used in this study (for example, contact information for the project steering committee) in your response to the relevant question in the submission form. See https://journals.plos.org/plosmedicine/s/data-availability for more information.

If the study has an associated protocol and/or analysis plan, please can these be provided as supplementary materials. 

Please add a completed STROBE checklist as a supplementary file, labelled "S1_STROBE_Checklist" or similar and referred to as such in the Methods section.

In the checklist, please refer to individual items by section (e.g., "Methods") and paragraph number, not by line or page numbers as these generally change in the event of publication.

Please update your in-text citations to appear before punctuation. 

The final paragraph of the ‘Introduction’ can be shortened to state the study’s aims only; elements of discussion can be relocated to the ‘Discussion’ section.

Please remove the ‘Transparency’ and ‘Conflict of Interest’ statements from the end of the main text. The relevant information should be captured in your responses to the submission forms and, in the event of publication, will be presented as metadata. 

Comments from the reviewers:

Reviewer #1: Thanks for the opportunity to review your manuscript. My role is as a statistical reviewer, so my comments and questions focus on the study design, data, and analysis presented in the manuscript. I have put general questions first, and then followed these with queries specific to a section of the manuscript (with a line/page reference).

This manuscript presents a study designed to identify key symptoms of COVID-19. This data is from community prevalence surveys in the UK where participants are recruited, questioned about symptom history and have a swab tested for COVID-19. The symptoms were identified by splitting surveys 2-7 into training and test data, and variable selection was achieved by using LASSO logistic regression with all the recorded symptoms. Within the test data sets, a resampling approach was used to estimate a stability of selection parameter, with variables selected 50% of the time were selected. This model was validated against the test set from surveys 2-7 (same temporality) and in survey 8. This is a complex but robust approach to variable selection with both LASSO and the resampling within training data. There was a fairly good level of performance in the round 8 data (AUC=0.77) given that the symptoms are all self-report.

There are some new approaches used here that I am not familiar with (i.e. particularly the approach to stability selection penalty parameter), is the code used for the analysis able to be shared? This would help me understand what you did and be able to fully recommend it. 

A pre-registered study protocol is available online for the overall REACT study. Was an analysis plan for this study created, and if so, is this available for the review? 

I have checked but I couldn't find what the overall recruitment rate for the REACT-1 study was. It looks as though the study uses quota sampling to meet the sample size objectives, how many potential participants at each wave were approached but didn't respond, relative to those that did participant in the study? (see question below about survey design and generalisability)

P7, L131. What software (R?) and packages were used to complete the analysis?

P7, L132. To clarify, the main analysis was repeated by COVID strain, age groups, and using first reported symptoms with each sub-group repeated separately from the others? 

P11, L235. Do you mean literally mean a 'predictive' model be applied in other contexts with the 7 symptoms or that policy around access to testing, isolation etc. should expand to include the extra symptoms? 

P11, L259. This is a good point - is there any data available from the overall survey that can show that this was achieved? I tried looking but couldn't find any.

P11, L263. 'Inaccurate' is very broad - what is a likely form of recall bias, and would induce spurious associations, or conceal real associations with this methodology? I have seen 'negative control' type of questions added to surveys e.g. symptoms that could not be associated with the disease included to see if there is an association with the outcome (although this might be hard given the range of symptoms COVID seems to bring about). 

Reviewer #2: This is an excellent study, methodologically well done, well written, and of public health importance, which leaves me little to say.

I have only a few suggestions:

(1) B117 was spreading in England from Sep 2020, and so would have been present in Round 5-7 samples. This confounds the comparison of Round 8 with earlier rounds, so please provide an estimate of what percentage of infections in England were likely B117 in earlier months, based on existing public domain estimates. It would have been interesting to show an additional analysis comparing Rounds 2-4 vs Round 8 for similar reasons, but this is not essential.

(2) P11/L234-236

This is a very important point, which perhaps could be stated more clearly for the reader. The important message here is that in countries where testing is being rationed by being based only on the first four symptoms, then almost certainly increasing the symptom list by the other 3 symptoms will be almost as efficient, whilst identifying a significantly larger number of infections.

(3) P12/L238-239

It's not clear to me what the intention was of making the reference to economically disadvantaging individuals. The issue of economic disadvantage at the individual level applies equally however restrictive the testing regime, since almost everyone who tests positive will be infected. I think the authors are trying to convey an economic point at the level of the population. But from an economic perspective, the best way to minimize the economic disadvantage of isolation must be to reduce the number of infected persons in the population, which is best achieved by reducing transmission (Reff) as much as possible. To achieve this, isolating a greater fraction of infected cases is always better, so increasing the number of individuals so identified in the short-term will always reduce the net number of people needing isolation in the longer-term.

(4) I refer to the following papers:

 Hellewell J, Abbott S, Gimma A, Bosse NI, Jarvis CI, Russell TW, et al. Feasibility of controlling COVID-19 outbreaks by isolation of cases and contacts. Lancet Glob Health. 2020;8(4):e488-e96.

Kerr et al, 2021, Nature Communications. Controlling COVID-19 via test-trace-quarantine. https://www.nature.com/articles/s41467-021-23276-9

I think efficiency is important in a context where testing capacity is limited, so this paper helps inform the testing policy in those contexts. However, to successfully control COVID-19, efficiency is not the only criterion. Effectiveness is also important, ie maximization of the fraction of all infections detected is critical to any strategy that aims to reduce R below 1, especially one that seeks to avoid damaging social mobility restrictions. I would suggest the authors discuss the implications for the less constrained settings, and scenarios where the primary goal is to stop transmission.

Specifically, the authors should discuss further their finding that the 7 symptoms would still only identify ~70% of symptomatic individuals, or 30-40% of all infected cases (after accounting for asymptomatic infections. Given that a large fraction of those with the relevant symptoms will not voluntarily test in the most ideal circumstances, what this means is that in practice the actual percentage of symptomatic cases that will be tested and detected will be far less. What the findings demonstrate is that the kind of strategy that Australia and New Zealand pursued makes a lot of sense. They used the full list of 26 symptoms to trigger testing (eg https://www.nsw.gov.au/covid-19/health-and-wellbeing/symptoms-and-testing). This does result in a much lower positivity rate than would be achieved by restricting to the more efficient subset of 7 symptoms (<0.1% vs 1%), but it achieves much better control of transmission (sufficient to achieve short-term elimination), and in the long run required less overall testing than in the UK (2 per 1000/d vs 20 per 1000/day). So for balance, the issue of (cost) efficiency needs to be considered in both the short term and long-term perspectives.

Reviewer #3: This study seeks to answer a policy-important question in a large representative population. This is a prevalence study using repeated cross-sectional surveys in a representative sample of the community in England aiming to identify a set of symptoms to predict SARS-CoV-2 PCR positivity and distinguish between wild type and B.1.1.7 strains. The surveys consisted of self-administered nose / throat swabs and questionnaires to a randomly selected sample. Significant symptoms were identified over 6 rounds of testing between June and December 2020 and were validated in a sub-population from these first 6 rounds and in an additional 7th round of testing in January 2021 when the B.1.1.7 strain was predominantly circulating. A large population of over 1 million people over the age of 5 were included. Seven symptoms were identified, and the authors have concluded that these may be more efficient than the current four symptoms recommended for testing eligibility within the UK national testing programme at present. This is one of the largest studies of this type that has been conducted and has monitored changes over the course of the pandemic.

Major issues

1. It would be useful to have a table or description of the included population (eg age, sex, region, occupations, socio-economic status, number of households) and number of participants per round of testing. 

Minor Issues

2.1 At what time point were surveys administered - before or after the swab? Did they relate to just the day that the swab was administered or to a period of time before sampling?

2.2 Were baseline questionnaires administered to allow chronic symptoms to be distinguished from acute?

2.3 How were the symptoms that were included in the questionnaire selected?

2.4 Were all swabs and viral isolates processed in the same laboratory? If not, is there a difference in assay performance that could affect the results?

2.5 There is a potential for selection bias due to survey response. Were there any differences between survey responders and non-responders?

2.6 There is no mention of whether sampling was carried out with or without replacement, although the study protocol states that it was with replacement. How were repeat samples in the same individuals accounted for?

2.7 Please describe any exclusion criteria if these were present. 

2.8 Please describe the sample size calculation. 

2.9 When comparing the wild type and the B.1.1.7 groups - were there any systematic differences (eg age) that could have introduced bias?

2.10 In discussion section, it would be useful to mention whether there have been other studies that have identified symptoms that can predict PCR positivity / developed risk scores and how this study compares to them.

[LINK]

---

## [Decision Letter · Decision Letter 2]

6 Aug 2021

Dear Dr. Chadeau-Hyam,

Thank you very much for re-submitting your manuscript "Predictive symptoms for COVID-19 in the community: REACT-1 study of over one million people" (PMEDICINE-D-21-02112R2) for review by PLOS Medicine.

I have discussed the paper with my colleagues and the academic editor and it was also seen again by three reviewers. I am pleased to say that provided the remaining editorial and production issues are dealt with we are planning to accept the paper for publication in the journal.

[LINK]

We look forward to receiving the revised manuscript by Aug 13 2021 11:59PM.   

Sincerely,

Callam Davidson, 

Associate Editor 

PLOS Medicine

plosmedicine.org

Requests from Editors:

Please add further detail to your data availability statement regarding how a reader could apply for access to the REACT study data via the project steering committee (a website or email address ought to be sufficient). 

Please reference the study questionnaires and the address at which they are located in your Methods section.

Please remove the ‘Contributors’, ‘Funding’, and ‘Data sharing statement’ from the end of the main text. In the event of publication, this information will be published as metadata based on your responses to the submission form. 

Please relocate the ‘Patient and public involvement’ and ‘Dissemination to participants and related patient and public communities’ sections to the Methods, rather than placing them at the end of the main text.

Citations should be in square brackets, not in superscript, and preceding punctuation.

Please use the "Vancouver" style for reference formatting, and see our website for other reference guidelines https://journals.plos.org/plosmedicine/s/submission-guidelines#loc-references

After updating your references to use Vancouver style, please ensure that references do not have periods after initials, use ‘et al.’ only after listing the first six authors, and do not contain any formatted text (e.g., bold, italics).

Please define any abbreviations used in figures (e.g., OR in Figures 3 and 4).

Comments from Reviewers:

Reviewer #1: Thank you for the revised manuscript and responses to my original queries. 

The additional explanation and available code for the Meinhausen and Buhlman stability selection approach were helpful thank you, I have also looked at the original reference and agree with you this is a an appropriate approach.

The rest of the clarifications to the manuscript address my original queries - the part of the discussion about generalisability and potential for bias looks better now. I agree with the comment about timing of self-report and availability of test results, and given the short time period between recall of symptoms and test I think the potential for bias here is limited.

My only query was about the analysis plan - on p2 of the sup information document there is the extra explanation of the stability selection approach and the link to the code, but I could not see an analysis plan or link to one. Have I inadvertently missed this? This could added to the supplementary appendix easily.

Great manuscript - I look forward to seeing whether symptoms change with delta when that data becomes available to you.

Reviewer #2: Thank you for the revisions which address the comments by me and other reviewers.

The only comment I would make is that in the last paragraph of the discussion, the use of the term "most efficient" is incorrect. Technically speaking, using the first ranked symptom would be the most efficient. What I think you mean is that the combination of seven symptoms identified would be the most optimal given the need to maximise case detection in the community under the constraints to testing capacity that existed in conditions similar to that in England in the relevant time period.

Ravi P. Rannan-Eliya

Reviewer #3: My comments have been addressed in this revised manuscript, thank you to the authors.

[LINK]

---

## [Editor Report · Decision Letter 3]

11 Aug 2021

Dear Dr. Chadeau-Hyam,

Thank you very much for re-submitting your manuscript "Predictive symptoms for COVID-19 in the community: REACT-1 study of over one million people" (PMEDICINE-D-21-02112R3) for review by PLOS Medicine.

There are still a small number of minor revisions that are required before we can proceed. The remaining issues that need to be addressed are listed at the end of this email. Please take these into account before resubmitting your manuscript.

In revising the manuscript for further consideration here, please ensure you address the specific points made by the editors. In your rebuttal letter you should indicate your response to the editors' comments and the changes you have made in the manuscript. Please submit a clean version of the paper as the main article file. A version with changes marked must also be uploaded as a marked up manuscript file.

We hope to receive your revised manuscript within 1 week. Please email me (cdavidson@plos.org) if you have any questions or concerns.

Sincerely,

Callam Davidson, 

Associate Editor 

PLOS Medicine

plosmedicine.org

Requests from Editors:

The data availability statement has changed since the last revision but not as described in your rebuttal letter – the previous revision read ‘Summary statistics and descriptive tables are available in Supplementary information; R scripts used to perform the analyses can be downloaded at https://github.com/barbarabodinier/Community_detection_COVID-19. REACT data can be accessed upon application and acceptance to the project steering committee.’ In the latest revision, the statement reads ‘Summary statistics and descriptive tables from REACT-1 study are available on the study website https://www.imperial.ac.uk/medicine/research-and-impact/groups/reactstudy/react-1-study-materials/ and, together with the R scripts used to perform the analyses, at https://github.com/barbarabodinier/Community_detection_COVID’. 

My issues with this are as follows:

* The summary statistics and descriptive tables are in the supplementary materials; the web address you have provided contains the questionnaires used. Please correct the statement to include both.

* It looks as though the statement has been truncated or text is missing, as ‘and, together’ does not scan properly and the github address is now missing ‘-19’ and leads to an error page. Please check carefully.

* The email addresses you mention having added in your rebuttal later are not available (in fact reference to the project steering committee has been removed altogether). Please check and see the comment below.

* Finally (and importantly), a study author cannot be the contact person for the data – if there is a contact within the project steering committee that is not a named author, or the option to use a data access or ethics committee, please provide their details. If you foresee this being a problem, please contact me (cdavidson@plos.org) to discuss.

Line 80: Please update this bullet point to ‘Data were collected from over 1 million participants in the REACT-1 study (June 2020 to January 2021), from whom 26 symptoms were assayed and the results of a PCR test were available.

Line 83: Please update this bullet point to ‘Adopting a variable selection approach, we sought to determine the best combination of symptoms jointly and complementarily predictive of PCR positivity and compared whether these symptoms were the same between individuals infected by the wild-type virus and those infected by the B.1.1.7 variant’. The following bullet point (line 86) can then be removed.

Line 89: Please update this bullet point to ‘We identify seven symptoms that were jointly predictive of PCR positivity and appeared to vary only marginally across age groups: loss or change of sense of smell, loss or change of sense of taste, fever, new persistent cough, chills, appetite loss and muscle aches’. The following bullet point (line 92) can then be removed.

Line 99: Please update ‘but will enable’ to ‘but would enable’ 

Line 455: Please remove the periods after the author initials in reference 15.

---

## [Editor Report · Decision Letter 4]

20 Aug 2021

Dear Dr Chadeau-Hyam, 

On behalf of my colleagues and the Academic Editor, Dr Kretzschmar, I am pleased to inform you that we have agreed to publish your manuscript "Predictive symptoms for COVID-19 in the community: REACT-1 study of over one million people" (PMEDICINE-D-21-02112R4) in PLOS Medicine.

PRESS

Sincerely, 

Richard Turner PhD, for Callam Davidson 

rturner@plos.org